# SeaFormer: Squeeze-enhanced Axial Transformer for Mobile Semantic Segmentation

**Qiang Wan**[1,*] **Zilong Huang**[2] **Jiachen Lu**[1] **Gang Yu**[2] **Li Zhang**[1,†]
[1]School of Data Science, Fudan University    [2]Tencent PCG

## Abstract

Since the introduction of Vision Transformers, the landscape of many computer vision tasks (*e.g.*, semantic segmentation), which has been overwhelmingly dominated by CNNs, recently has significantly revolutionized. However, the computational cost and memory requirement render these methods unsuitable on the mobile device, especially for the high-resolution per-pixel semantic segmentation task. In this paper, we introduce a new method *squeeze-enhanced Axial Transformer* (***SeaFormer***) for mobile semantic segmentation. Specifically, we design a generic attention block characterized by the formulation of squeeze Axial and detail enhancement. It can be further used to create a family of backbone architectures with superior cost-effectiveness. Coupled with a light segmentation head, we achieve the best trade-off between segmentation accuracy and latency on the ARM-based mobile devices on the ADE20K and Cityscapes datasets. Critically, we beat both the mobile-friendly rivals and Transformer-based counterparts with better performance and lower latency without bells and whistles. Beyond semantic segmentation, we further apply the proposed SeaFormer architecture to image classification problem, demonstrating the potentials of serving as a versatile mobile-friendly backbone. Our code and models are made publicly available at `https://github.com/fudan-zvg/SeaFormer`.

## 1 Introduction

As a fundamental problem in computer vision, semantic segmentation aims to assign a semantic class label to each pixel in an image. Conventional methods rely on stacking local convolution kernel Long et al. (2015) to perceive the long-range structure information of the image.

Since the introduction of Vision Transformers Dosovitskiy et al. (2021), the landscape of semantic segmentation has significantly revolutionized. Transformer-based approaches Zheng et al. (2021); Xie et al. (2021) have remarkably demonstrated the capability of global context modeling. However, the computational cost and memory requirement of Transformer render these methods unsuitable on mobile devices, especially for high-resolution imagery inputs.

Following conventional wisdom of efficient operation, local/window-based attention Luong et al. (2015); Liu et al. (2021); Huang et al. (2021a); Yuan et al. (2021), Axial attention Huang et al. (2019b); Ho et al. (2019); Wang et al. (2020a), dynamic graph message passing Zhang et al. (2020; 2022b) and some lightweight attention mechanisms Hou et al. (2020); Li et al. (2021b;c; 2020); Liu et al. (2018); Shen et al. (2021); Xu et al. (2021); Cao et al. (2019); Woo et al. (2018); Wang et al. (2020b); Choromanski et al. (2021); Chen et al. (2017); Mehta & Rastegari (2022a) are introduced.

However, these advances are still insufficient to satisfy the design requirements and constraints for mobile devices due to the high latency on the high-resolution inputs (see Figure 1). Recently there is a surge of interest in building a Transformer-based semantic segmentation. In order to reduce the computation cost at high resolution, TopFormer Zhang et al. (2022c) dedicates to applying the global attention at a $1/64$ scale of the original input, which definitely harms the segmentation performance.

To solve the dilemma of high-resolution computation for pixel-wise segmentation task and low latency requirement on the mobile device in a performance harmless way, we propose a family

---

*Work done while Qiang Wan was an intern at Tencent GY-Lab.
†Corresponding author: lizhangfd@fudan.edu.cn

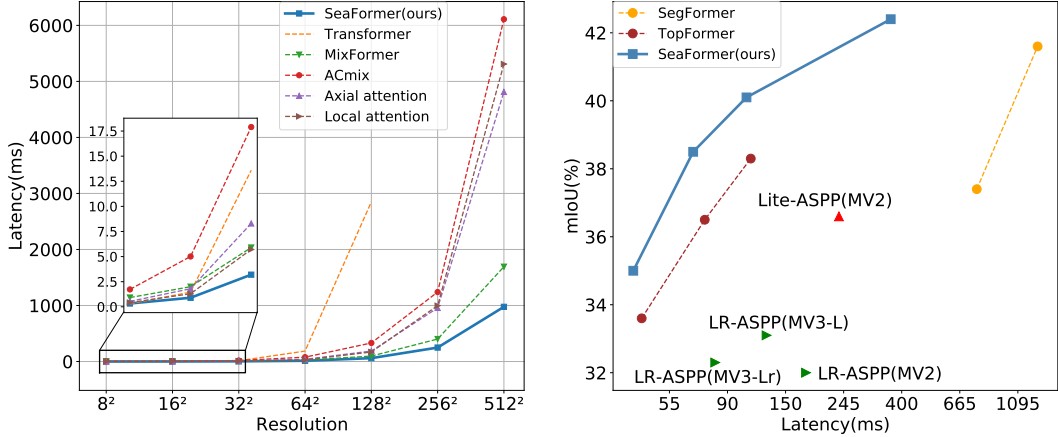

Figure 1: **Left**: Latency comparison with Transformer Vaswani et al. (2017), MixFormer Chen et al. (2022a), ACmix Pan et al. (2022b), Axial attention Ho et al. (2019) and local attention Luong et al. (2015). It is measured with a single module of channel dimension 64 on a Qualcomm Snapdragon 865 processor. **Right**: The mIoU versus latency on the ADE20K *val* set. MV2 means MobileNetV2 Sandler et al. (2018). MV3-L means MobileNetV3-Large Howard et al. (2019). MV3-Lr denotes MobileNetV3-Large-reduce Howard et al. (2019). The latency is measured on a single Qualcomm Snapdragon 865, and only an ARM CPU core is used for speed testing. No other means of acceleration, e.g., GPU or quantification, is used. For figure *Right*, the input size is 512×512. SeaFormer achieves superior trade-off between mIoU and latency.

of mobile-friendly Transformer-based semantic segmentation model, dubbed as *squeeze-enhanced Axial Transformer* (SeaFormer), which reduces the computational complexity of axial attention from $\mathcal{O}((H+W)HW)$ to $\mathcal{O}(HW)$, to achieve superior accuracy-efficiency trade-off on mobile devices and fill the vacancy of mobile-friendly efficient Transformer.

The core building block *squeeze-enhanced Axial attention* (SEA attention) seeks to squeeze (pool) the input feature maps along the horizontal/vertical axis into a compact column/row and computes self-attention. We concatenate query, keys and values to compensate the detail information sacrificed during squeeze and then feed it into a depth-wise convolution layer to enhance local details.

Coupled with a light segmentation head, our design (see Figure 2) with the proposed SeaFormer layer in the small-scale feature is capable of conducting high-resolution image semantic segmentation with low latency on the mobile device. As shown in Figure 1, the proposed SeaFormer outperforms other efficient neural networks on the ADE20K dataset with lower latency. In particular, SeaFormer-Base is superior to the lightweight CNN counterpart MobileNetV3 (41.0 *vs.* 33.1 mIoU) with lower latency (106ms *vs.* 126ms) on an ARM-based mobile device.

We make the following **contributions**: **(i)** We introduce a novel *squeeze-enhanced Axial Transformer* (SeaFormer) framework for mobile semantic segmentation; **(ii)** Critically, we design a generic attention block characterized by the formulation of squeeze Axial and detail enhancement; It can be used to create a family of backbone architectures with superior cost-effectiveness; **(iii)** We show top performance on the ADE20K and Cityscapes datasets, beating both the mobile-friendly rival and Transformer-based segmentation model with clear margins; **(iv)** Beyond semantic segmentation, we further apply the proposed SeaFormer architecture to the image classification problem, demonstrating the potential of serving as a versatile mobile-friendly backbone.

## 2  RELATED WORK

**Combination of Transformers and convolution**   Convolution is relatively efficient but not suitable to capture long-range dependencies and vision Transformer has the powerful capability with a global receptive field but lacks efficiency due to the computation of self-attention. In order to make full use of both of their advantages, MobileViT Mehta & Rastegari (2022a), TopFormer Zhang et al. (2022c), LVT Yang et al. (2022), Mobile-Former Chen et al. (2022b), EdgeViTs Pan et al. (2022a), MobileViTv2 Mehta & Rastegari (2022b), EdgeFormer Zhang et al. (2022a) and EfficientFormer Li et al. (2022) are constructed as efficient ViTs by combining convolution with Transformers. Mobile-

ViT, Mobile-Former, TopFormer and EfficientFormer are restricted by Transformer blocks and have to trade off between efficiency and performance in model design. LVT, MobileViTv2 and EdgeViTs keep the model size small at the cost of relatively high computation, which also means high latency.

**Axial attention and variants**   Axial attention Huang et al. (2019b); Ho et al. (2019); Wang et al. (2020a) is designed to reduce the computational complexity of original global self-attention Vaswani et al. (2017). It computes self-attention over a single axis at a time and stacks a horizontal and a vertical axial attention module to obtain the global receptive field. Strip pooling Hou et al. (2020) and Coordinate attention Hou et al. (2021) uses a band shape pooling window to pool along either the horizontal or the vertical dimension to gather long-range context. Kronecker Attention Networks Gao et al. (2020) uses the juxtaposition of horizontal and lateral average matrices to average the input matrices and performs attention operation. These methods and other similar implementations provide performance gains partly at considerably low computational cost compared with Axial attention. However, they ignore the lack of local details brought by the pooling/average operation.

**Mobile semantic segmentation**   The mainstream of efficient segmentation methods are based on lightweight CNNs. DFANet Li et al. (2019) adopts a lightweight backbone to reduce computation cost and adds a feature aggregation module to refine high-level and low-level features. ICNet Zhao et al. (2018) designs an image cascade network to speed up the algorithm, while BiSeNet Yu et al. (2018; 2021) proposes two-stream paths for low-level details and high-level context information, separately. Fast-SCNN Poudel et al. (2019) shares the computational cost of the multi-branch network to yield a run-time fast segmentation CNN. TopFormer Zhang et al. (2022c) presents a new architecture with a combination of CNNs and ViT and achieves a good trade-off between accuracy and computational cost for mobile semantic segmentation. However, it is still restricted by the heavy computation load of global self-attention.

## 3   METHOD

### 3.1   OVERALL ARCHITECTURE

Inspired by the two-branch architectures Yu et al. (2021); Poudel et al. (2019); Hong et al. (2021); Huang et al. (2021b); Chen et al. (2022b), we design a **s**queeze-**e**nhanced **A**xial Trans**former** (**SeaFormer**) framework. As is shown in Figure 2, SeaFormer consists of these parts: *shared STEM*, *context branch*, *spatial branch*, *fusion block* and *light segmentation head*. For a fair comparison, we follow TopFormer Zhang et al. (2022c) to design the STEM. It consists of one regular convolution with stride of 2 followed by four MobileNet blocks where stride of the first and third block is 2. The context branch and the spatial branch share the produced feature map, which allows us to build a fast semantic segmentation model.

**Context branch**   The context branch is designed to capture context-rich information from the feature map $\mathbf{x}_s$. As illustrated in the red branch of Figure 2, the context branch is divided into three stages. To obtain larger receptive field, we stack SeaFormer layers after applying a MobileNet block to down-sampling and expanding feature dimension. Compared with the standard convolution as the down-sampling module, MobileNet block increases the representation capacity of the model while maintaining a lower amount of computation and latency. For variants except SeaFormer-Large, SeaFormer layers are applied in the last two stages for superior trade-off between accuracy and efficiency. For SeaFormer-Large, we insert SeaFormer layers in each stage of context branch. To achieve a good trade-off between segmentation accuracy and inference speed, we design a squeeze-enhanced Axial attention block (SEA attention) illustrated in the next subsection.

**Spatial branch**   The spatial branch is designed to obtain spatial information in high resolution. Identical to the context branch, the spatial branch reuses feature maps $\mathbf{x}_s$. However, the feature from the early convolution layers contains rich spatial details but lacks high-level semantic information. Consequently, we design a fusion block to fuse the features in the context branch into the spatial branch, bringing high-level semantic information into the low-level spatial information.

**Fusion block**   As depicted in Figure 2, high resolution feature maps in the spatial branch are followed by a $1 \times 1$ convolution and a batch normalization layer to produce a feature to fuse. Low

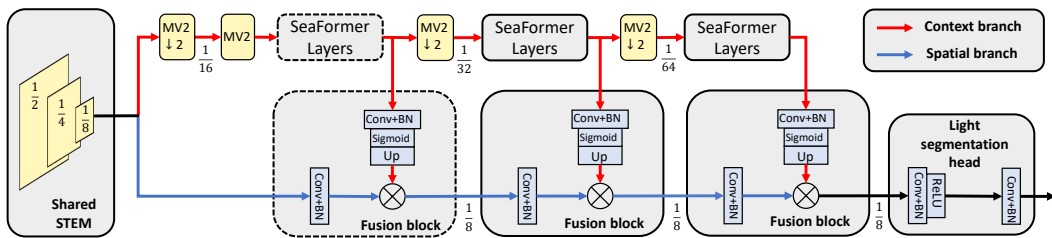

Figure 2: The overall architecture of SeaFormer. It contains shared STEM, context branch (**red**), spatial branch (**blue**), fusion block and light segmentation head. `MV2` block means MobileNetV2 block and `MV2 ↓2` means MobileNetV2 block with downsampling. SeaFormer layers and fusion block with dash box only exist in SeaFormer-L. The symbol $\otimes$ denotes element-wise multiplication.

resolution feature maps in the context branch are fed into a $1 \times 1$ convolution layer, a batch normalization layer, a sigmoid layer and up-sampled to high resolution to produce semantics weights by bilinear interpolation. Then, the semantics weights from context branch are element-wisely multiplied to the high resolution feature from spatial branch. The fusion block enables low-level spatial features to obtain high-level semantic information.

**Light segmentation head**  The feature after the last fusion block is fed into the proposed segmentation head directly, as demonstrated in Figure 2. For fast inference purpose, our light segmentation head consists of two convolution layers, which are followed by a batch normalization layer separately and the feature from the first batch normalization layer is fed into an activation layer.

## 3.2 SQUEEZE-ENHANCED AXIAL ATTENTION

The global attention can be expressed as

$$
\mathbf{y}_o = \sum_{p \in \mathcal{G}(o)} \text{softmax}_p \left( \mathbf{q}_o^\top \mathbf{k}_p \right) \mathbf{v}_p \tag{1}
$$

where $\mathbf{x} \in \mathbb{R}^{H \times W \times C}$. $\mathbf{q}, \mathbf{k}, \mathbf{v}$ are linear projection of $\mathbf{x}$, $i.e. \mathbf{q} = \mathbf{W}_q \mathbf{x}, \mathbf{k} = \mathbf{W}_k \mathbf{x}, \mathbf{v} = \mathbf{W}_v \mathbf{x}$, where $\mathbf{W}_q, \mathbf{W}_k \in \mathbb{R}^{C_{qk} \times C}, \mathbf{W}_v \in \mathbb{R}^{C_v \times C}$ are learnable weights. $\mathcal{G}(o)$ means all positions on the feature map of location $o = (i, j)$. When traditional attention module is applied on a feature map of $H \times W \times C$, the time complexity can be $\mathcal{O}(H^2 W^2 (C_{qk} + C_v))$, leading to low efficiency and high latency.

$$
\mathbf{y}_o = \sum_{p \in \mathcal{N}_{m \times m}(o)} \text{softmax}_p \left( \mathbf{q}_o^\top \mathbf{k}_p \right) \mathbf{v}_p \tag{2}
$$

$$
\mathbf{y}_o = \sum_{p \in \mathcal{N}_{1 \times W}(o)} \text{softmax}_p \left( \mathbf{q}_o^\top \mathbf{k}_p \right) \mathbf{v}_p + \sum_{p \in \mathcal{N}_{H \times 1}(o)} \text{softmax}_p \left( \mathbf{q}_o^\top \mathbf{k}_p \right) \mathbf{v}_p \tag{3}
$$

To improve the efficiency, there are some works Liu et al. (2021); Huang et al. (2019b); Ho et al. (2019) computing self-attention within the local region. We show two most representative efficient Transformer in Equation 2, 3. Equation 2 is represented by window-based attention Luong et al. (2015) successfully reducing the time complexity to $\mathcal{O}(m^2 HW (C_{qk} + C_v)) = \mathcal{O}(HW)$, where $\mathcal{N}_{m \times m}(o)$ means the neighbour $m \times m$ positions of $o$, but loosing global receptiveness. The Equation 3 is represented by Axial attention Ho et al. (2019), which only reduces the time complexity to $\mathcal{O}((H + W) HW (C_{qk} + C_v)) = \mathcal{O}((HW)^{1.5})$, where $\mathcal{N}_{H \times 1}(o)$ means all the positions of the column of $o$; $\mathcal{N}_{1 \times W}(o)$ means all the positions of the row of $o$.

According to their drawbacks, we propose the mobile-friendly squeeze-enhanced Axial attention, with a succinct squeeze Axial attention for global semantics extraction and an efficient convolution-based detail enhancement kernel for local details supplement.

$$
\mathbf{q}_{(h)} = \frac{1}{W} \left( \mathbf{q}^{\to (C_{qk}, H, W)} \mathbb{1}_W \right)^{\to (H, C_{qk})}, \quad \mathbf{q}_{(v)} = \frac{1}{H} \left( \mathbf{q}^{\to (C_{qk}, W, H)} \mathbb{1}_H \right)^{\to (W, C_{qk})} \tag{4}
$$

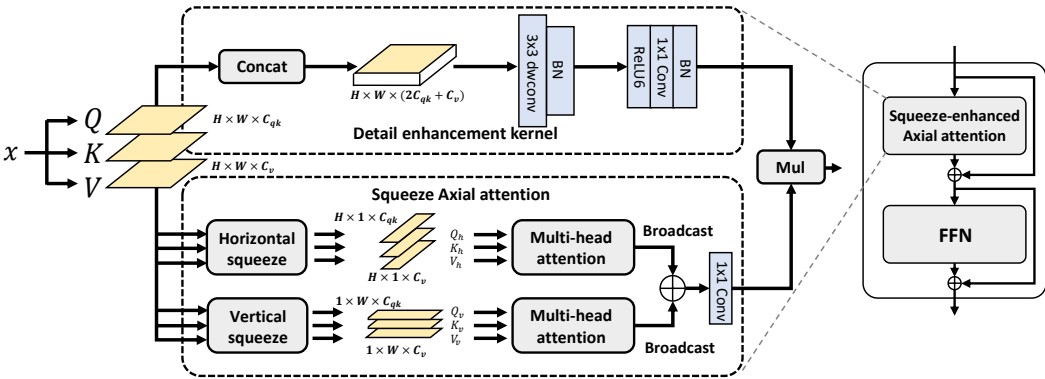

Figure 3: **Right**: the schematic illustration of the proposed squeeze-enhanced Axial Transformer layer including a squeeze-enhanced Axial attention and a Feed-Forward Network (FFN). **Left** is the squeeze-enhanced Axial Transformer layer, including detail enhancement kernel and squeeze Axial attention. The symbol $\bigoplus$ indicates an element-wise addition operation. Mul means multiplication.

**Squeeze Axial attention**   To achieve a more efficient computation and aggregate global information at the same time, we resort to a more radical strategy. In the same way, $\mathbf{q}, \mathbf{k}, \mathbf{v}$ are first get from $\mathbf{x}$ with $\mathbf{W}_q^{(s)}, \mathbf{W}_k^{(s)} \in \mathbb{R}^{C_{qk} \times C}, \mathbf{W}_v^{(s)} \in \mathbb{R}^{C_v \times C}$. According to Equation 4, we first implement *horizontal squeeze* by taking average of query feature map on the horizontal direction. In the same way, the right shows the *vertical squeeze* on the vertical direction. $\mathbf{z}^{\rightarrow(\cdot)}$ means permuting the dimension of tensor $\mathbf{z}$ as given, and $\mathbb{1}_m \in \mathbb{R}^m$ is a vector with all the elements equal to 1. The squeeze operation on $\mathbf{q}$ also repeats on $\mathbf{k}$ and $\mathbf{v}$, so we finally get $\mathbf{q}_{(h)}, \mathbf{k}_{(h)}, \mathbf{v}_{(h)} \in \mathbb{R}^{H \times C_{qk}}$, $\mathbf{q}_{(v)}, \mathbf{k}_{(v)}, \mathbf{v}_{(v)} \in \mathbb{R}^{W \times C_{qk}}$. The squeeze operation reserves the global information to a single axis, thus greatly alleviating the following global semantic extraction showing by Equation 5.

$$\mathbf{y}_{(i,j)} = \sum_{p=1}^{H} \text{softmax}_p \left( \mathbf{q}_{(h)i}^{\top} \mathbf{k}_{(h)p} \right) \mathbf{v}_{(h)p} + \sum_{p=1}^{W} \text{softmax}_p \left( \mathbf{q}_{(v)j}^{\top} \mathbf{k}_{(v)p} \right) \mathbf{v}_{(v)p} \tag{5}$$

Each position of feature map propagates information only on two squeezed axial features. Although it shows no distinct computation reduction comparing to Equation 3, repeat of Equation 5 can be simply implemented by the most efficient `broadcast` operation. The detail is shown in Figure 3. Time complexity for squeezing $\mathbf{q}, \mathbf{k}, \mathbf{v}$ is $\mathcal{O}((H + W)(2C_{qk} + C_v))$ and the attention operation takes $\mathcal{O}((H^2 + W^2)(C_{qk} + C_v))$ time. Thus, our squeeze Axial attention successfully reduces time complexity to $\mathcal{O}(HW)$.

**Squeeze Axial position embedding**   Equation 4 are, however, not positional-aware, including no positional information of feature map. Hence, we propose squeeze Axial position embedding to squeeze Axial attention. For squeeze Axial attention, we render both $\mathbf{q}_{(h)}$ and $\mathbf{k}_{(h)}$ to be aware of their position in squeezed axial feature by introducing positional embedding $\mathbf{r}_{(h)}^q, \mathbf{r}_{(h)}^k \in \mathbb{R}^{H \times C_{qk}}$, which are linearly interpolated from learnable parameters $\mathbf{B}_{(h)}^q, \mathbf{B}_{(h)}^k \in \mathbb{R}^{L \times C_{qk}}$. $L$ is a constant. In the same way, $\mathbf{r}_{(v)}^q, \mathbf{r}_{(v)}^k \in \mathbb{R}^{W \times C_{qk}}$ are applied to $\mathbf{q}_{(v)}, \mathbf{k}_{(v)}$. Thus, the positional-aware squeeze Axial attention can be expressed as Equation 6.

$$\mathbf{y}_{(i,j)} = \sum_{p=1}^{H} \text{softmax}_p \left( (\mathbf{q}_{(h)i} + \mathbf{r}_{(h)i}^q)^{\top} (\mathbf{k}_{(h)p} + \mathbf{r}_{(h)p}^k) \right) \mathbf{v}_{(h)p}$$
$$+ \sum_{p=1}^{W} \text{softmax}_p \left( (\mathbf{q}_{(v)j} + \mathbf{r}_{(v)j}^q)^{\top} (\mathbf{k}_{(v)p} + \mathbf{r}_{(v)p}^k) \right) \mathbf{v}_{(v)p} \tag{6}$$

**Detail enhancement kernel**   The squeeze operation, though extracting global semantic information efficiently, sacrifices the local details. Hence an auxiliary convolution-based kernel is applied to enhance the spatial details. As is shown in the upper path of Figure 3, $\mathbf{q}, \mathbf{k}, \mathbf{v}$ are first get from $\mathbf{x}$ with another $\mathbf{W}_q^{(e)}, \mathbf{W}_k^{(e)} \in \mathbb{R}^{C_{qk} \times C}, \mathbf{W}_v^{(e)} \in \mathbb{R}^{C_v \times C}$ and are concatenated on the channel dimension and then passed to a block made up of 3×3 depth-wise convolution and batch normalization.

| Backbone | Decoder | Params | FLOPs | mIoU | Latency |
|---|---|---|---|---|---|
| MobileNetV2 | LR-ASPP | 2.2M | 2.8G | 32.0 | 177ms |
| MobileNetV3-Large-reduce | LR-ASPP | 1.6M | 1.3G | 32.3 | 81ms |
| MobileNetV3-Large | LR-ASPP | 3.2M | 2.0G | 33.1 | 126ms |
| HRNet-W18-Small | HRNet-W18-Small | 4.0M | 10.2G | 33.4 | 639ms |
| TopFormer-T | Simple Head | 1.4M | 0.6G | 33.6 | 43ms |
| TopFormer-T* | Simple Head | 1.4M | 0.6G | 34.6 | 43ms |
| **SeaFormer-T** | Light Head | 1.7M | 0.6G | **35.0** | **40ms** |
| **SeaFormer-T*** | Light Head | 1.7M | 0.6G | **35.8±0.35** | **40ms** |
| ConvMLP-S | Semantic FPN | 12.8M | 33.8G | 35.8 | 777ms |
| EfficientNet | DeepLabV3+ | 17.1M | 26.9G | 36.2 | 970ms |
| MobileNetV2 | Lite-ASPP | 2.9M | 4.4G | 36.6 | 235ms |
| TopFormer-S | Simple Head | 3.1M | 1.2G | 36.5 | 74ms |
| TopFormer-S* | Simple Head | 3.1M | 1.2G | 37.0 | 74ms |
| **SeaFormer-S** | Light Head | 4.0M | 1.1G | **38.1** | **67ms** |
| **SeaFormer-S*** | Light Head | 4.0M | 1.1G | **39.4±0.25** | **67ms** |
| MiT-B0 | SegFormer | 3.8M | 8.4G | 37.4 | 770ms |
| ResNet18 | Lite-ASPP | 12.5M | 19.2G | 37.5 | 648ms |
| ShuffleNetV2-1.5x | DeepLabV3+ | 16.9M | 15.3G | 37.6 | 960ms |
| MobileNetV2 | DeepLabV3+ | 15.4M | 25.8G | 38.1 | 1035ms |
| TopFormer-B | Simple Head | 5.1M | 1.8G | 38.3 | 110ms |
| TopFormer-B* | Simple Head | 5.1M | 1.8G | 39.2 | 110ms |
| **SeaFormer-B** | Light Head | 8.6M | 1.8G | **40.2** | **106ms** |
| **SeaFormer-B*** | Light Head | 8.6M | 1.8G | **41.0±0.45** | **106ms** |
| MiT-B1 | SegFormer | 13.7M | 15.9G | 41.6 | 1300ms |
| **SeaFormer-L** | Light Head | 14.0M | 6.5G | **42.7** | **367ms** |
| **SeaFormer-L*** | Light Head | 14.0M | 6.5G | **43.7±0.36** | **367ms** |

Table 1: Results of semantic segmentation on ADE20K *val* set, * indicates training batch size is 32. The latency is measured on a single Qualcomm Snapdragon 865 with input size 512×512, and only an ARM CPU core is used for speed testing. References: MobileNetV2 Sandler et al. (2018), MobileNetV3 Howard et al. (2019), HRNet Yuan et al. (2020), TopFormer Zhang et al. (2022c), ConvMLP Li et al. (2021a), Semantic FPN Kirillov et al. (2019), EfficientNet Tan & Le (2019), DeepLabV3+ and Lite-ASPP Chen et al. (2018a), SegFormer Xie et al. (2021), ResNet He et al. (2016), ShuffleNetV2-1.5x Ma et al. (2018).

By using a 3×3 convolution, auxiliary local details can be aggregated from $\mathbf{q}, \mathbf{k}, \mathbf{v}$. And then a linear projection with activation function and batch normalization are used to squeeze $(2C_{qk} + C_v)$ dimension to $C$ and generate detail enhancement weights. Finally, the enhancement feature will be fused with the feature given by squeeze Axial attention. Different enhancement mode including element-wise addition and multiplication will be compared in experiment section. Time complexity for the 3×3 depth-wise convolution is $\mathcal{O}(3^2 HW(2C_{qk} + C_v))$ and the time complexity for the 1×1 convolution is $\mathcal{O}(HWC(2C_{qk} + C_v))$. Time for the other operations like activation can be omitted.

**Architecture and Variants** We introduce four variants, SeaFormer-Tiny, Small, Base and Large (T, S, B and L). More configuration details are listed in the supplementary material.

## 4 EXPERIMENTS

We evaluate our method on semantic segmentation and image classification tasks. First, we describe implementation details and compare results with state of the art. We then conduct a series of ablation studies to validate the design of SeaFormer. Each proposed component and important hyper-parameters are examined thoroughly.

### 4.1 EXPERIMENTAL SETUP

#### 4.1.1 DATASET

We perform segmentation experiments over ADE20K Zhou et al. (2017), CityScapes Cordts et al. (2016). The mean of intersection over union (mIoU) is set as the evaluation metric. We convert full-precision models to TNN Contributors (2019) and measure latency on an ARM-based device with a single Qualcomm Snapdragon 865 processor.

| Method | Backbone | FLOPs | mIoU(val) | mIoU(test) | Latency |
|--------|----------|-------|-----------|------------|---------|
| FCN | MobileNetV2 | 317G | 61.5 | - | 24190ms |
| PSPNet | MobileNetV2 | 423G | 70.2 | - | 31440ms |
| SegFormer(h) | MiT-B0 | 17.7G | 71.9 | - | 1586ms |
| SegFormer(f) | MiT-B0 | 125.5G | 76.2 | - | 11030ms |
| L-ASPP | MobileNetV2 | 12.6G | 72.7 | - | 887ms |
| LR-ASPP | MobileNetV3-L | 9.7G | 72.4 | 72.6 | 660ms |
| LR-ASPP | MobileNetV3-S | 2.9G | 68.4 | 69.4 | 211ms |
| Simple Head(h) | TopFormer-B | 2.7G | 70.7 | - | 173ms |
| Simple Head(f) | TopFormer-B | 11.2G | 75.0 | 75.0 | 749ms |
| Light Head(h) | **SeaFormer-S** | 2.0G | 70.7 | 71.0 | 129ms |
| Light Head(f) | **SeaFormer-S** | 8.0G | 76.1 | 75.9 | 518ms |
| Light Head(h) | **SeaFormer-B** | 3.4G | 72.2 | 72.5 | 205ms |
| Light Head(f) | **SeaFormer-B** | 13.7G | 77.7 | 77.5 | 821ms |

Table 2: Results on Cityscapes *val* set. The results on *test* set of some methods are not presented due to the fact that they are not reported in their original papers.

**ADE20K** dataset covers 150 categories, containing 25K images that are split into 20K/2K/3K for *Train*, *val* and *test*. **CityScapes** is a driving dataset for semantic segmentation. It consists of 5000 fine annotated high-resolution images with 19 categories.

### 4.1.2 IMPLEMENTATION DETAILS

We set ImageNet-1K Deng et al. (2009) pretrained network as the backbone, and training details of ImageNet-1K are illustrated in the last subsection. For semantic segmentation, the standard Batch-Norm Ioffe & Szegedy (2015) layer is replaced by synchronized BatchNorm.

**Training**  Our implementation is based on public codebase `mmsegmentation` Contributors (2020). We follow the batch size, training iteration scheduler and data augmentation strategy of TopFormer Zhang et al. (2022c) for a fair comparison. The initial learning rate is 0.0005 and the weight decay is 0.01. A "poly" learning rate scheduled with factor 1.0 is adopted. During inference, we set the same resize and crop rules as TopFormer to ensure fairness. The comparison of Cityscapes contains full-resolution and half-resolution. For the full-resolution version, the training images are randomly scaled and then cropped to the fixed size of $1024 \times 1024$. For the half-resolution version, the training images are resized to $1024 \times 512$ and randomly scaling, the crop size is $1024 \times 512$.

### 4.2 COMPARISON WITH STATE OF THE ART

**ADE20K**  Table 1 shows the results of SeaFormer and previous efficient backbones on ADE20K *val* set. The comparison covers Params, FLOPs, Latency and mIoU. As shown in Table 1, SeaFormer outperforms these approaches with comparable or less FLOPs and lower latency. Compared with specially designed mobile backbone, TopFormer, which sets global attention as its semantics extractor, SeaFormer achieves higher segmentation accuracy with lower latency. And the performance of SeaFormer-B surpasses MobileNetV3 by a large margin of +7.9% mIoU with lower latency (-16%). The results demonstrate our SeaFormer layers improve the representation capability significantly.

**Cityscapes**  From the table 2, it can be seen that SeaFormer-S achieves comparable or better results than TopFormer-B with less computation cost and latency, which proves that SeaFormer could also achieve a good trade-off between performance and latency in high-resolution scenario.

### 4.3 ABLATION STUDIES

In this section, we ablate different self-attention implementations and some important design elements in the proposed model, including our squeeze-enhanced Axial attention module (SEA attention) and fusion block on ADE20K dataset.

**The influence of components in SEA attention**  We conduct experiments with several configurations, including detail enhancement kernel only, squeeze Axial attention only, and the fusion of both. As is shown in Table 3, only detail enhancement or squeeze Axial attention achieves a relatively poor

| Enhance Attn kernel | branch | Enhance input | Enhance mode | Params | FLOPs | Latency | Top1 | mIoU |
|---|---|---|---|---|---|---|---|---|
| ✔ | | - | - | 1.3M | 0.58G | 38ms | 65.9 | 32.5 |
| | ✔ | - | - | 1.4M | 0.57G | 38ms | 66.3 | 33.5 |
| ✔ | ✔ | conv(x) | Mul | 1.6M | 0.60G | 40ms | 67.2 | 34.9 |
| ✔ | ✔ | upconv(x) | Mul | 1.8M | 0.62G | 41ms | 68.1 | 35.9 |
| ✔ | ✔ | concat[qkv] | Mul | 1.7M | 0.60G | 40ms | 67.9 | 35.8 |
| ✔ | ✔ | concat[qkv] | Add | 1.7M | 0.60G | 40ms | 67.3 | 35.4 |

Table 3: Ablation studies on components in SEA attention on ImageNet-1K and ADE20K datasets. Enhancement input means the input of detail enhancement kernel. conv(x) means x followed by a point-wise conv. upconv(x) is the same as conv(x) except different channels as upconv(x) is from $C_{in}$ to $C_q + C_k + C_v$ and conv(x) is from $C_{in}$ to $C_{in}$. concat[qkv] indicates concat of Q,K, V.

performance, and enhancing squeeze Axial attention with detail enhancement kernel brings a performance boost with a gain of 2.3% mIoU on ADE20K. The results indicate that enhancing global semantic features from squeeze Axial attention with local details from convolution optimizes the feature extraction capability of Transformer block. For enhancement input, there is an apparent performance gap between upconv(x) and conv(x). And we conclude that increasing the channels will boost performance significantly. Comparing concat[qkv] and upconv(x), which also correspond to w/ or w/o convolution weight sharing between detail enhancement kernel and squeeze Axial attention, we can find that sharing weights makes our model improve inference efficiency with minimal performance loss (35.8 *vs.*35.9). As for enhancement modes, multiplying features from squeeze Axial attention and detail enhancement kernel outperforms add enhancement by +0.4% mIoU.

**Comparison with different self-attention modules** In order to eliminate the impact of our architecture and demonstrate the effectiveness and generalization ability of SEA attention, we ran experiments on Swin Transformer Liu et al. (2021) by replacing window attention in Swin Transformer with different attention blocks. We set the same training protocol, hyper-parameters, and model architecture configurations as Swin for a fair comparison. When replacing window attention with CCAttention (CCNet) or DoubleAttention (A2-Nets), they have much lower FLOPs than SeaFormer

| Model | Params(B) | FLOPs(B) | mIoU | Latency |
|---|---|---|---|---|
| Swin | 27.5M | 25.6G | 44.5 | 3182ms |
| CCNet | 41.6M | 37.4G | 43.1 | 3460ms |
| ISSA | 31.8M | 33.3G | 37.4 | 2991ms |
| A2-Nets | 37.2M | 31.1G | 28.9 | 2502ms |
| Axial | 36.2M | 32.5G | 45.3 | 3121ms |
| Local | 27.5M | 25.1G | 34.2 | 3059ms |
| MixFormer | 27.5M | 24.9G | 45.5 | 2817ms |
| ACmix | 27.9M | 26.6G | 45.3 | 3712ms |
| Global | 27.5M | 0.144T | OOM | 14642ms |
| **SeaFormer** | 34.0M | **24.9G** | **46.5** | **2278ms** |

Table 4: Results on ADE20K *val* set based on Swin Transformer architecture. (B) denotes backbone. OOM means CUDA out of memory. References: ISSA Huang et al. (2019a), A2-Nets Chen et al. (2018b)

and other attention blocks. Considering that we may not be able to draw conclusions rigorously, we doubled the number of their Transformer blocks (including MLP). As ACmix has the same architecture configuration as Swin, we borrow the results from the original paper. From Table 4, it can be seen that SeaFormer outperforms other attention mechanisms with lower FLOPs and latency.

**The influence of the width in fusion block** To study the influence of the width in fusion block, we perform experiments with different embedding dimensions in fusion blocks on SeaFormer-Base, M denotes the channels that spatial branch and context branch features mapping to in two fusion blocks. Results are shown in Table 5.

## 4.4 IMAGE CLASSIFICATION

We conduct experiments on ImageNet-1K Deng et al. (2009), which contains 1.28M training

images and 50K validation images from 1,000 classes. We employ an AdamW Kingma & Ba (2014) optimizer for 600 epochs using a cosine decay learning rate scheduler. A batch size of 1024, an initial learning rate of 0.064, and a weight decay of 2e-5 are used. The results are illustrated in Table 6. Compared with other efficient approaches, SeaFormer achieves a relatively better trade-off between latency and accuracy.

| M | Params | FLOPs | Latency | mIoU |
|---|---|---|---|---|
| 64,96 | 8.5M | 1.7G | 102ms | 40.3 |
| 128,160 | 8.6M | 1.8G | 106ms | 41.0 |
| 192,256 | 8.7M | 2.0G | 121ms | 41.2 |

| Posbias | Params | FLOPs | Latency | mIoU |
|---|---|---|---|---|
| ✗ | 1.65M | 0.60G | 40ms | 35.6 |
| ✔ | 1.67M | 0.60G | 40ms | 35.8 |

Table 5: Ablation studies on embedding dimensions and position bias. M = [128, 160] is an optimal embedding dimension in fusion blocks.

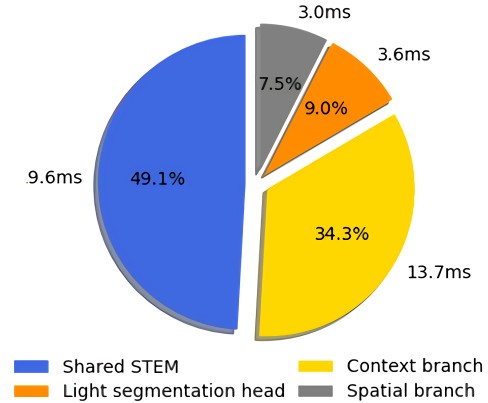

Figure 4: The inference latency of components.

### 4.5 LATENCY STATISTICS

We make the statistics of the latency of the proposed SeaFormer-Tiny, as shown in Figure 4, the shared STEM takes up about half of the latency of the whole network (49%). The latency of the context branch is about a third of the total latency (34%), whilst the actual latency of the spatial branch is relatively low (8%) due to sharing early convolution layers with the context branch. Our light segmentation head (8%) also contributes to the success of building a light model.

| Method | P(M) | F(G) | Top1 | L |
|---|---|---|---|---|
| MobileNetV3-Small | 2.9 | 0.1 | 67.4 | **5ms** |
| **SeaFormer-T** | 1.8 | 0.1 | **67.9** | 7ms |
| MobileViT-XXS | 1.3 | 0.4 | 69.0 | 24ms |
| MobileViTv2-0.50 | 1.4 | 0.5 | 70.2 | 32ms |
| MobileOne-S0* | 2.1 | 0.3 | 71.4 | 14ms |
| MobileNetV2 | 3.4 | 0.3 | 72.0 | 17ms |
| Mobile-Former96 | 4.8 | 0.1 | 72.8 | 31ms |
| **SeaFormer-S** | 4.1 | 0.2 | **73.3** | **12ms** |
| EdgeViT-XXS | 4.1 | 0.6 | 74.4 | 71ms |
| LVT | 5.5 | 0.9 | 74.8 | 97ms |
| MobileViT-XS | 2.3 | 0.9 | 74.8 | 54ms |
| MobileNetV3-Large | 5.4 | 0.2 | 75.2 | **16ms** |
| Mobile-Former151 | 7.7 | 0.2 | 75.2 | 42ms |
| MobileViTv2-0.75 | 2.9 | 1.0 | 75.6 | 68ms |
| MobileOne-S1* | 4.8 | 0.8 | 75.9 | 40ms |
| **SeaFormer-B** | 8.7 | 0.3 | **76.0** | 20ms |
| MobileOne-S2* | 7.8 | 1.3 | 77.4 | 63ms |
| EdgeViT-XS | 6.8 | 1.1 | 77.5 | 124ms |
| MobileViTv2-1.00 | 4.9 | 1.8 | 78.1 | 115ms |
| MobileOne-S3* | 10.1 | 1.9 | 78.1 | 91ms |
| MobileViT-S | 5.6 | 1.8 | 78.4 | 88ms |
| EfficientNet-B1 | 7.8 | 0.7 | 79.1 | 61ms |
| EfficientFormer-L1 | 12.3 | 1.3 | 79.2 | 94ms |
| Mobile-Former508 | 14.8 | 0.5 | 79.3 | 102ms |
| MobileOne-S4* | 14.8 | 3.0 | 79.4 | 143ms |
| **SeaFormer-L** | 14.0 | 1.2 | **79.9** | **61ms** |

Table 6: Image classification results on ImageNet-1K *val* set. The FLOPs and latency are measured with input size 224×224, except for MobileViT and MobileViTv2 that are measured with 256×256 according to their original implementations. `P`, `F` and `L` mean `Parameters`, `FLOPs` and `latency`. * indicates re-parameterized variants Vasu et al. (2022). The latency is measured on a single Qualcomm Snapdragon 865, and only an ARM CPU core is used for speed testing. No other means of acceleration, e.g., GPU or quantification, is used.

## 5 CONCLUSION

In this paper, we have proposed *squeeze-enhanced Axial Transformer* (**SeaFormer**) for mobile semantic segmentation, filling the vacancy of mobile-friendly efficient Transformer. Moreover, we create a family of backbone architectures of SeaFormer and achieve cost-effectiveness. The superior performance on the ADE20K and Cityscapes, and the lowest latency demonstrate its effectiveness on the ARM-based mobile device. Beyond semantic segmentation, we further apply the proposed SeaFormer architecture to image classification problem, demonstrating the potential of serving as a versatile mobile-friendly backbone.

### ACKNOWLEDGMENTS

This work was supported in part by National Natural Science Foundation of China (Grant No. 62106050), Lingang Laboratory (Grant No. LG-QS-202202-07), Natural Science Foundation of Shanghai (Grant No. 22ZR1407500) and CCF-Tencent Open Research Fund (No. CCF-Tencent RAGR20210111).

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

# Appendix

## A   ARCHITECTURE DETAILS AND VARIANTS

SeaFormer backbone contains 6 stages, corresponding to the shared STEM and context branch in Figure 2 in the main paper. When conducting the image classification experiments, a pooling layer and a linear layer are added at the end of the context branch.

Table 7 details the family of our SeaFormer configurations with varying capacities. We construct SeaFormer-Tiny, SeaFormer-Small, SeaFormer-Base and SeaFormer-Large models with different scales via varying the number of SeaFormer layers and the feature dimensions. We use input image size of $512 \times 512$ by default. For variants except SeaFormer-Large, SeaFormer layers are applied in the last two stages for superior trade-off between accuracy and efficiency. For SeaFormer-Large, we apply the proposed SeaFormer layers in each stage of the context branch.

## B   COMPLEXITY ANALYSIS

we analyze the complexity of our proposed SEA attention in subsection 3.2 to demonstrate its efficiency theoretically. In our application, we set $C_{qk} = 0.5C_v$ to further reduce computation cost. The total time complexity of squeeze-enhanced Axial attention is

$$
\begin{aligned}
&\mathcal{O}((H^2 + W^2)(C_{qk} + C_v)) + \mathcal{O}((H + W)(2C_{qk} + C_v)) + \mathcal{O}((HWC + 9HW)(2C_{qk} + C_v)) \\
&= \mathcal{O}((1.5H^2 + 1.5W^2 + 2HWC + 18HW + 2H + 2W)C_v) = \mathcal{O}(HW),
\end{aligned} \tag{7}
$$

if we assume $H = W$ and take channel as constant. SEA attention is linear to the feature map size theoretically. Moreover, SEA attention only includes mobile-friendly operation like convolution, pooling, matrix multiplication and so on.

## C   PASCAL CONTEXT PERFORMANCE

We evaluate performance on Pascal Context *val* set over 59 categories and 60 categories. **PASCAL Context** dataset has 4998/5105 images for *train* and *test*, covering 59 semantic labels and 1 background.

Following TopFormer Zhang et al. (2022c), we train the models for 80,000 iterations on PASCAL Context dataset. The same data augmentation strategy and batch size are adopted for a fair comparison. The initial learning rate is 0.0002 and the weight decay is 0.01. A poly learning rare scheduled with factor 1.0 is used.

Table 8 demonstrates that SeaFormer-S is +1.4% mIoU higher (45.08% vs.43.68%) than TopFormer-S with lower latency.

## D   COCO-STUFF PERFORMANCE

We compare SeaFormer with the previous approaches on COCO-Stuff *val* set. **COCO-Stuff** dataset augments COCO dataset with pixel-level stuff annotations. 10K complex images are selected from COCO. The *train* and *test* set contain 9K/1K images.

Following TopFormer Zhang et al. (2022c), we train the models for 80,000 iterations on COCO-Stuff dataset. The same data augmentation strategy and batch size are adopted for a fair comparison. The initial learning rate is 0.0002 and the weight decay is 0.01. A poly learning rare scheduled with factor 1.0 is used.

Table 9 reveals that SeaFormer-S is +2.0% mIoU higher (32.82% vs.30.83%) than TopFormer-S with less computation cost and lower latency.

| | Resolution | SeaFormer-Tiny | SeaFormer-Small | SeaFormer-Base | SeaFormer-Large |
|---|---|---|---|---|---|
| Stage1 | H/2 × W/2 | [Conv, 3, 16, 2]
[MB, 3, 1, 16, 1] | [Conv, 3, 16, 2]
[MB, 3, 1, 16, 1] | [Conv, 3, 16, 2]
[MB, 3, 1, 16, 1] | [Conv, 3, 32, 2]
[MB, 3, 3, 32, 1] |
| Stage2 | H/4 × W/4 | [MB, 3, 4, 16, 2]
[MB, 3, 3, 16, 1] | [MB, 3, 4, 24, 2]
[MB, 3, 3, 24, 1] | [MB, 3, 4, 32, 2]
[MB, 3, 3, 32, 1] | [MB, 3, 4, 64, 2]
[MB, 3, 4, 64, 1] |
| Stage3 | H/8 × W/8 | [MB, 5, 3, 32, 2]
[MB, 5, 3, 32, 1] | [MB, 5, 3, 48, 2]
[MB, 5, 3, 48, 1] | [MB, 5, 3, 64, 2]
[MB, 5, 3, 64, 1] | [MB, 5, 4, 128, 2]
[MB, 5, 4, 128, 1] |
| Stage4 | H/16 × W/16 | [MB, 3, 3, 64, 2]
[MB, 3, 3, 64, 1] | [MB, 3, 3, 96, 2]
[MB, 3, 3, 96, 1] | [MB, 3, 3, 128, 2]
[MB, 3, 3, 128, 1] | [MB, 3, 4, 192, 2]
[MB, 3, 4, 192, 1]
[Sea, 3, 8] |
| Stage5 | H/32 × W/32 | [MB, 5, 3, 128, 2]
[Sea, 2, 4] | [MB, 5, 4, 160, 2]
[Sea, 3, 6] | [MB, 5, 4, 192, 2]
[Sea, 4, 8] | [MB, 5, 4, 256, 2]
[Sea, 3, 8] |
| Stage6 | H/64 × W/64 | [MB, 3, 6, 160, 2]
[Sea, 2, 4] | [MB, 3, 6, 192, 2]
[Sea, 3, 6] | [MB, 3, 6, 256, 2]
[Sea, 4, 8] | [MB, 3, 6, 320, 2]
[Sea, 3, 8] |

Table 7: Architectures for semantic segmentation. [Conv, 3 ,16, 2] denotes regular convolution layer with kernel of 3, output channel of 16 and stride of 2. [MB, 3, 4, 16, 2] means MobileNetV2 Sandler et al. (2018) block with kernel of 3, expansion ratio of 4, output channel of 16 and stride of 2. [Sea, 2, 4] refers to SeaFormer layers with number of layers of 2 and heads of 4.

| Backbone | Decoder | F(G) | mIoU(60/59) |
|---|---|---|---|
| MBV2-s16 | DeepLabV3+ | 22.24 | 38.59/42.34 |
| ENet-s16 | DeepLabV3+ | 23.00 | 39.19/43.07 |
| MBV3-s16 | LR-ASPP | 2.04 | 35.05/38.02 |
| TopFormer-T | Simple Head | 0.53 | 36.41/40.39 |
| **SeaFormer-T** | Light Head | 0.51 | **37.27/41.49** |
| TopFormer-S | Simple Head | 0.98 | 39.06/43.68 |
| **SeaFormer-S** | Light Head | 0.98 | **40.20/45.08** |
| TopFormer-B | Simple Head | 1.54 | 41.01/45.28 |
| **SeaFormer-B** | Light Head | 1.57 | **41.77/45.92** |

Table 8: Results on Pascal Context *val* set. `F` means `FLOPs`. We omit the latency as the input resolution is almost the same as that in table 1.

| Backbone | Decoder | F(G) | mIoU |
|---|---|---|---|
| MBV2-s8 | PSPNet | 52.94 | 30.14 |
| ENet-s16 | DeepLabV3+ | 27.10 | 31.45 |
| MBV3-s16 | LR-ASPP | 2.37 | 25.16 |
| TopFormer-T | Simple Head | 0.64 | 28.34 |
| **SeaFormer-T** | Light Head | 0.62 | **29.24** |
| TopFormer-S | Simple Head | 1.18 | 30.83 |
| **SeaFormer-S** | Light Head | 1.15 | **32.82** |
| TopFormer-B | Simple Head | 1.83 | 33.43 |
| **SeaFormer-B** | Light Head | 1.81 | **34.07** |

Table 9: Results on COCO-Stuff *test* set. `F` means `FLOPs`. We omit the latency in this table as the input resolution is the same as that in table 1.

| Backbone | AP | FLOPs | Params |
|---|---|---|---|
| ShuffleNetv2 Ma et al. (2018) | 25.9 | 161G | 10.4M |
| **SeaFormer-T** | **31.5** | **160G** | 10.9M |
| MF151 | 34.2 | 161G | 14.4M |
| MV3 | 27.2 | 162G | 12.3M |
| **SeaFormer-S** | **34.6** | **161G** | 13.3M |
| MF214 | 35.8 | **162G** | 15.2M |
| MF294 | 36.6 | 164G | 16.1M |
| **SeaFormer-B** | **36.7** | 164G | 18.1M |
| ResNet50 He et al. (2016) | 36.5 | 239G | 37.7M |
| PVT-Tiny Wang et al. (2021) | 36.7 | 221G | 23.0M |
| ConT-M Yan et al. (2021) | 37.9 | 217G | 27.0M |
| **SeaFormer-L** | **39.8** | **185G** | 24.0M |

Table 10: Results on COCO object detecion. MF denotes Mobile-Former Chen et al. (2022b). MV3 denotes MobileNetV3 Howard et al. (2019).

| Fusion method | mIoU |
|---|---|
| Add directly | 35.2 |
| Multiply directly | 35.2 |
| Sigmoid add | 34.8 |
| **Sigmoid multiply** | **35.8** |

Table 11: Ablation study on fusion method on ADE20K *val* set.

# E  OBJECT DETECTION PERFORMANCE

To further demonstrate the generalization ability of our proposed SeaFormer backbone on other downstream tasks, we conduct object detection task on COCO dataset.

**Setup**  We use RetinaNet Lin et al. (2017) (one-stage) as the detection framework and follow the standard settings to use SeaFormer as backbone to generate e feature pyramid at multiple scales. All models are trained on train2017 split for 12 epochs (1×) from ImageNet pretrained weights.

**Results**  From the table 10 We can observe that our SeaFormer achieves superior results on detection task which further demonstrates the strong generalization ability of our method.

# F  ADDITIONAL ABLATION STUDY

In addition to the ablation study in the submission paper, we investigate the effect of fusion method in fusion block in Figure 2.

## F.1  THE INFLUENCE OF FUSION BLOCK DESIGN

We set four fusion methods, including "Add directly", "Multiply directly", "Sigmoid add" and "Sigmoid multiply". **X** directly means features from context branch and spatial branch **X** directly. Sigmoid **X** means feature from context branch goes through a sigmoid layer and **X** feature from spatial branch.

From the Table 11 we can see that replacing sigmoid multiply with other fusion methods hurts performance. Sigmoid multiply is our optimal fusion block choice.

## F.2  EFFECTIVE AND EFFICIENCY OF SEA ATTENTION

To verify the effectiveness and efficiency of SEA attention based on our designed pipeline, we experiment with convolution, Global attention, Local attention, Axial attention and three convolution

| Method | Params | FLOPs | Latency | Top1 | mIoU |
|---|---|---|---|---|---|
| Conv | 1.6M | 0.59G | 38ms | 66.3 | 32.8 |
| Local | 1.3M | 0.60G | 48ms | 65.9 | 32.8 |
| Axial | 1.6M | 0.63G | 44ms | 66.9 | 33.7 |
| Global | 1.3M | 0.61G | 43ms | 66.7 | 34.2 |
| ACmix | 1.3M | 0.60G | 54ms | 66.0 | 33.1 |
| MixFormer | 1.3M | 0.60G | 50ms | 66.8 | 33.8 |
| **SeaFormer** | 1.7M | 0.60G | **40ms** | **67.9** | **35.8** |

Table 12: Performance of different self-attention modules on our designed pipeline on ImageNet-1K and ADE20K datasets.

| Model | mIoU | FP32 | FP16 |
|---|---|---|---|
| TopFormer-T | 34.6 | 43ms | 23ms |
| **SeaFormer-T** | **35.8** | **40ms** | **22ms** |
| TopFormer-S | 37.0 | 74ms | 41ms |
| **SeaFormer-S** | **39.4** | **67ms** | **36ms** |
| TopFormer-B | 39.2 | 110ms | 60ms |
| **SeaFormer-B** | **41.0** | **106ms** | **56ms** |
| SeaFormer-L | 43.7 | 367ms | 186ms |

Table 13: Performance comparison on ADE20K *val* set under different precision.

enhanced attention methods including our SEA attention, ACmix and MixFormer. The ablation experiments are organized in seven groups. Since the resolution of computing attention is relatively small, the window size in Local attention, ACmix, and MixFormer is set to 4. We adjust the channels when applying different attention modules to keep the FLOPs aligned and compare their performance and latency. The results are illustrated in Table 12.

As demonstrated in the table, SEA attention outperforms the counterpart built on other efficient attentions. Compared with global attention, SEA attention outperforms it by +1.2% Top1 accuracy on ImageNet-1K and +1.6 mIoU on ADE20K with less FLOPs and lower latency. Compared with similar convolution enhanced attention works, ACmix and MixFormer, our SEA attention obtains better results on ImageNet-1K and ADE20K with similar FLOPs but lower latency. The results indicate the effectiveness and efficiency of SEA attention module.

# G  PERFORMANCE UNDER DIFFERENT PRECISION OF THE MODELS

Following TopFormer, we measure the latency in the submission paper on a single Qualcomm Snapdragon 865, and only an ARM CPU core is used for speed testing. No other means of acceleration, e.g., GPU or quantification, is used. We provide a more comprehensive comparison to demonstrate the necessity of our proposed method. We test the latency under different precision of the models. From the table 13, it can be seen that whether it is full precision or half precision the performance of SeaFormer is better than that of TopFormer.

# H  VISUALIZATION

## H.1  ATTENTION HEATMAP

To demonstrate the effectiveness of detail enhancement in our squeeze-enhanced Axial attention (SEA attention), we ablate our model by removing the detail enhancement. We visualize the attention heatmaps of the two models in Figure 5. Without detail enhancement, attention heatmaps from solely SA attention appears to be axial strips while our proposed SEA attention is able to activate the semantic local region accurately, which is particularly significant in the dense prediction task.

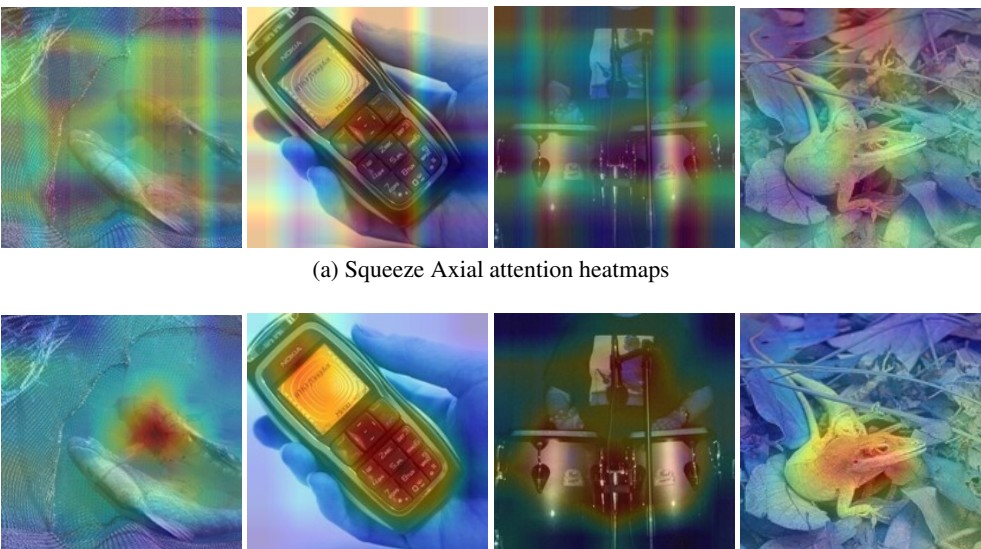

(a) Squeeze Axial attention heatmaps

(b) Squeeze-enhanced Axial attention heatmaps

Figure 5: The visualization of attention heatmaps from the model consisting of squeeze Axial attention without detail enhancement (*first row*) and SeaFormer (*second row*). Heatmaps are produced by averaging channels of the features from the last attention block, normalizing to [0, 255] and up-sampling to the image size.

## H.2   PREDICTION RESULTS

We show the qualitative results and compare with the alternatives on the ADE20K validation set from two different perspectives. First we compare with a mobile-friendly rival TopFormer Zhang et al. (2022c) with similar FLOPs and latency in Figure 6. Besides, we compare with the Transformer-based counterpart SegFormer-B1 Xie et al. (2021) in Figure 7. In particular, our SeaFormer-L has lower computation cost than the SegFormer-B1. As shown in both figures, we demonstrate better segmentation results than both the mobile counterpart and Transformer-based approach.

## I   LIMITATIONS AND SOCIETAL IMPACT

The mobile-friendly segmentation is deeply related to the industrial application on edge computation platforms, while few academic attempts are made to meet the requirement of the industry. We test our method on a Qualcomm Snapdragon 865 processor (Fig.1 main paper) and shows superior results to the alternatives. We believe our work can lead to expected and unexpected innovations in both academia and industry.

However, our system is not perfect yet and hence not fully trustworthy in real-world deployment. Also, the current system is not exhaustively evaluated and tested due to limited resources. We focus on mobile semantic segmentation and image classification tasks. New mobile-friendly method for more downstream tasks and extended to GPU systems will be studied in the future.

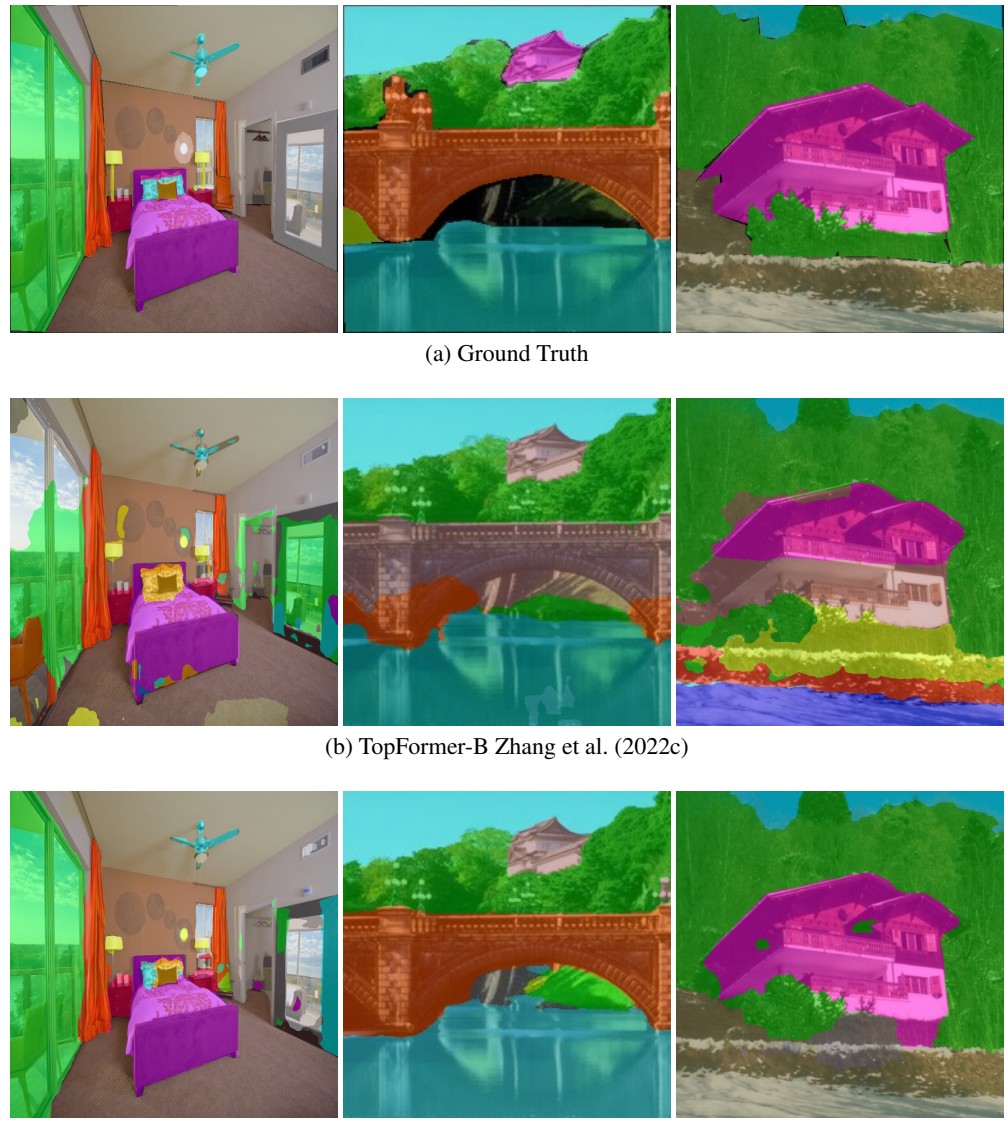

(a) Ground Truth

(b) TopFormer-B Zhang et al. (2022c)

(c) SeaFormer-B (Ours)

Figure 6: Visualization of prediction results on ADE20K *val* set.

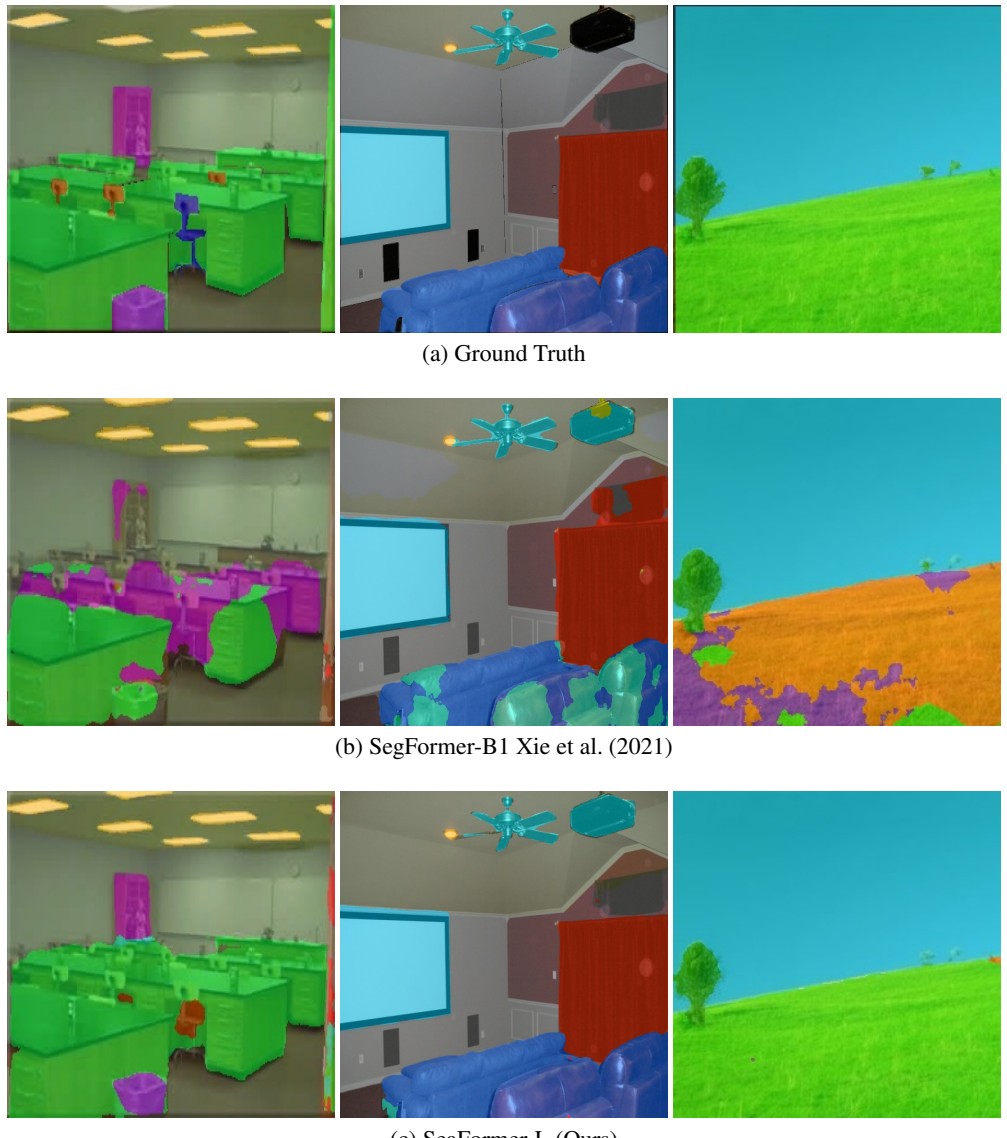

(a) Ground Truth

(b) SegFormer-B1 Xie et al. (2021)

(c) SeaFormer-L (Ours)

Figure 7: Visualization of prediction results on ADE20K *val* set.

