# OpenReview forum: "SeaFormer: Squeeze-enhanced Axial Transformer for Mobile Semantic Segmentation"
_ICLR.cc/2023/Conference — ICLR 2023 poster_

### Official Review · Reviewer_6wEd · 2022-10-24

**Confidence:** 2
**Correctness:** 4
**Technical Novelty And Significance:** 3
**Empirical Novelty And Significance:** 3
**Recommendation:** 8

**Clarity, Quality, Novelty And Reproducibility:**

This paper is clear, of good quality, with contribution and associated code based on openmmlab is provided.

**Strength And Weaknesses:**

Strength:
- The proposed transformer is O(WH), which breaks the original drawback of transformers. Even if many existing models already proposed modification in order to do similar optimisation, this paper combines an Axial attention branch (context branch) with a high resolution convolution branch (spatial branch). Both experiments and ablation study show that this is a relevant strategy.
- Both experiments and ablation study are convincing for semantic segmentation.

Weaknesses:
- The main contribution seems to be the new transformer architecture combining context and spatial branches, resulting to a generic bloc that breaks the complexity. This contribution is mainly presented in the context of mobile segmentation application. I wonder if considering in a more generic way this contribution would implies more impact: 1) As you point with the small experiment on classification, this is very generic and can be used for classification, detection, segmentation (semantic, instances, ...); and 2) blocks are designed for mobile applications (MobileNet blocks) and could be extended to GPU systems with more classical operations without changing the O(WH) complexity. Extending experiments in this way would be interesting with may be more impact for the paper.

**Summary Of The Paper:**

this paper presents SeaFormer: a new transformer architecture designed to reduce the well-known complexity problem with the vanilla structure. The resulting architecture, O(HW), is applied in a semantic segmentation task for mobile devices. Experiments shows SOTA results in a latency/mIoU. An ablation study shows the impact of each proposed contribution An additional short experiment is proposed to show that this new transformer architecture is generic and can be used in a classification task.

**Summary Of The Review:**

The main contribution of the paper is a new attention block with O(WH) complexity, which combines an axial attention branch with a high resolution convolutional block. This block is used into an semantic segmentation task designed for mobile devices. Experiments are convincing. As I pointed before, it should be interesting to extend experiments in order to show the genericity of the approach regarding other tasks and GPU systems.

---

> ### Author Response · Authors · 2022-11-16
> **Response to Reviewer 6wEd**
>
> Thanks for the positive and detailed review as well as the suggestions for improvement. We would like to reply to the comments as follows:
>
> **Q: A generic block**
>
> A: Thanks for the valuable suggestion. We have included the experiments of object detection on COCO in the supplementary material. For convenience, we paste the results below. It shows that our method outperforms the alternative methods by a clear margin, demonstrating the potential of serving as a versatile mobile-friendly backbone. We will add more discussion for this direction in the revision. In addition to the arm-based devices, we will add more experiments on GPU devices in the revision.
>
> |Backbone| AP| FLOPs| Params|
> |------------|---|--------|-----------|
> |ShuffleNetv2  |25.9 |161G |10.4M|
> |SeaFormer-T |31.5 |160G|10.9M|
> |MobileFormer151| 34.2 |161G |14.4M|
> |MobileNetV3 |27.2 |162G |12.3M|
> |SeaFormer-S |34.6 |161G |13.3M|
> |MobileFormer214 |35.8 |162G |15.2M|
> |MobileFormer294 |36.6 |164G |16.1M|
> |SeaFormer-B |36.7 |164G |18.1M|
> |ResNet50  |36.5 |239G |37.7M|
> |PVT-Tiny  |36.7 |221G |23.0M|
> |ConT-M  |37.9 |217G |27.0M|
> |SeaFormer-L |39.8 |185G |24.0M|

---

### Official Review · Reviewer_wqtJ · 2022-10-24

**Confidence:** 4
**Correctness:** 3
**Technical Novelty And Significance:** 2
**Empirical Novelty And Significance:** 2
**Recommendation:** 3

**Clarity, Quality, Novelty And Reproducibility:**

Clarity and Quality:
The paper is kind of easy to read, but the writing may be a little careless, such as citations, notation in Tables and figures.

Novelty:
The novelty of this paper is not satisfactory, since both pipeline and proposed modules are not proposed at the first time. (Discussed in Weakness)

Reproducibility:
The paper provides code.


**Details Of Ethics Concerns:**

No ethics concern.

**Strength And Weaknesses:**

Strength:
+ The squeeze axial attention saves a lot of cost, enabling real-time performance.
+ To overcome the loss of details in the global context extraction, there is a design for detail enhancement using depth-wise convolution to aggregate local spatial details.
+ The experiments show gains from TopFormer.

Weakness
- The pipeline of the proposed method is most built on TopFormer, which also have dual branches of spatial feature pyramid and global semantics, and fusion blocks. The only difference is the axial attention and detail enhancement kernel. However, the axial attention is not novel, and was utilized in previous segmentation works, e.g., Axial-DeepLab, Max-DeepLab. What’s the difference between the proposed axial attention and previous works? For the detail enhancement kernel, is it in the SeaFormer layer in context branch or spatial branch? The Fusion block in Figure 3 is the same as the one in Figure 2?
- Although the proposed model have about 1% mIoU gain over TopFormer, it is not the state-of-the-art. It is not suitable to claim this method achieves the state-of-the-art performance.
- A lot of citations are from arXiv, but some are already published.
- For the results on Cityscape dataset in supplemental material, the latency seems higher than TopFormer with same backbone. Why not present results of the half resolution or more backbones for more comparison?
- In table 3, it may be better to present the latency of components (although found in Supplemental material).
- Better to separate Table 2, Table 4. No (a, b, c) annotation for Table 4. Table 4 is referenced before Table 3.
- The number of parameters of the SeaFormer is about 1.2 ~1.6 times of TopFormer’s. What are the additional cost?


**Summary Of The Paper:**

This paper presents a model called squeeze-enhanced Axial Transformer (SeaFormer) for mobile semantic segmentation. The model is based on the previous TopFormer with two-branch architecture and integrates axial attention and enhancement into an attention block. The proposed attention block reduced memory and time complexity a lot. The proposed model achieved encouraging performance on several semantic segmentation datasets, i.e., ADE20K, Pascal Context and COCO-stuff.

**Summary Of The Review:**

This paper presents a model based on previous work with incremental achievement. The proposed axial attention and enhancement is not novel. The evaluation of the important datasets, cityscape, is not clear. Thus, I consider this paper may not be ready to be published in ICLR.

---

> ### Author Response · Authors · 2022-11-17
> **Response to Reviewer wqtJ [Part 2/2]**
>
> **Q4: Detail enhancement kernel and fusion block.**
>
> The detail enhancement kernel is a part of SeaFormer layer in the context path. \
> The fusion in Figure 3 is different from that of Figure 2. The fusion operation in Figure 3 is the mode of fusing information from detail enhancement kernel and squeeze axial attention, which is represented by add or multiply operation. We have ablated it in Table 3 (page 8). The fusion block in Figure 2 consists of convolution and batch normalization, as illustrated in the figure clearly. We have revised the name of the fusion in Figure 3 to avoid confusion in the revision.
>
> **Q5: The claim of state-of-the-art.**
>
> Thanks. In terms of accuracy alone, the proposed method does not achieve state-of-the-art performance. We have revised the statement to make it more clear that the proposed method achieves the best trade-off between segmentation accuracy and latency on mobile devices.
>
> **Q6: Citation format.**
>
> Thanks, we have updated the bib file in the revision.
>
> **Q7: Cityscapes results.**
>
> As shown in Table 2 (page 7), SeaFormer-B is 2.5 points better than TopFormer-B with only a slight increase in latency, showing the benefit of our efficient architecture design with multiple SeaFormer layers embedded in.\
> It is worth noting that with less computation cost and latency, our SeaFormer-Small even outperforms TopFormer-Base. This result further confirms the performance and efficiency of our model when processing high-resolution input images.
>
> **Q8: Results of the half resolution.**
>
> We have presented the results of the half resolution for SegFormer, TopFormer, and our SeaFormer on the CityScapes dataset (Table 2, page 8). Note that some of the results are not presented in this Table due to the fact that they are not reported in the original paper.
>
> **Q9: Results of more backbones.**
>
> Apart from the whole backbone architecture we designed, we have additionally validated the effectiveness of the proposed SeaFormer layer in the Swin Transformer architecture, showing that our proposed SeaFormer layer generalizes very well to multiple backbone architectures. Please see Table 2 of the supplementary materials. We have put this comparison in Table 4 of the revised paper based on reviewer xU22's suggestion.\
> In terms of the backbone we compared, we have included results with backbones including MobileNetV2, MobileNetV3, SegFormer and TopFormer on CityScapes dataset (Table 2 of the revised main paper) and MobileNetV2, MobileNetV3, HRNet, ConvMLP, EfficientNet, ShuffleNet, ResNet, SegFormer and TopFormer on ADE20K dataset (Table 1).
>
> **Q10: The latency of components found in the supplementary.**
>
> Thanks for the advice. We have moved the latency statistics of components into the main paper.
>
> **Q11: Table separation.**
>
> Thanks for the good suggestion. We have separated these tables and revised the order in the revision.
>
> **Q12: Parameters cost.**
>
> The additional parameters cost lies in two aspects. i) SEA attention has more parameters than global attention in TopFormer. ii) SeaFormer layers are applied in the last two or three stages, which is different from TopFormer that only uses Transformer layers in the last stage due to the high latency of the global attention.

---

> ### Author Response · Authors · 2022-11-17
> **Response to Reviewer wqtJ [Part 1/2]**
>
> Thanks for the insightful and detailed review as well as the suggestions for improvement. Our response to the comments is below:
>
> **Q1: Novelty.**
>
> 1) **The intuition behind our work is that we need to solve the dilemma of high-resolution computation for pixel-wise segmentation task and low latency requirement on mobile devices.** This is non-trivial and we thus propose a generic attention block characterized by the formulation of squeeze Axial and spatial enhancement and further create a family of backbone architectures with superior cost-effectiveness to beat state-of-the-art alternatives.
> 2) **We strongly advocate that our approach per se significantly revolutionizes mobile semantic segmentation**, especially for the high-resolution per-pixel semantic segmentation task. The other virtue is that our model also tackles the image classification and object detection problem, demonstrating the potential of serving as a versatile mobile-friendly backbone.
> 3) Regarding performance, we beat both the state-of-the-art mobile-friendly rivals and state-of-the-art Transformer-based counterparts with better performance and lower latency without bells and whistles.
> 4) Given its conceptual simplicity, versatility, and fast speed, our method can serve as a strong baseline and inspire further studies that consider mobile-friendly methods on edge computation platforms. We believe our work can lead to expected and unexpected innovations in both academia and industry.
>
> **Q2: Difference between the proposed axial attention and previous works.**
>
> 1) Our contribution lies in the proposed single attention module, **but more importantly** in the whole backbone architecture design. We strongly advocate that **the latter** very much matters for the mobile-friendly task.
> 2) Designing a single attention module and integrating it into existing backbone architectures is actually **cheap**. The importance of efficiency is usually underscored by the fact that the capabilities of many mobile and embedded devices are limited. This is also why most of the mobile-friendly works usually focus on the whole architecture design (eg, MobileNet).
> 3) Specifically, the proposed squeeze-enhanced Axial attention consists of two parts: squeeze axial attention and detail enhancement kernel. It is different from that of the previous works, e.g., Axial-DeepLab, Max-DeepLab which use the original axial attention. The difference lies in four aspects, i) **Squeeze Axial Attention v.s. Axial Attention.** For squeeze axial attention, the input feature will be average pooling/squeezed along horizontal (width) or vertical (height) dimensions, before being fed into the self-attention module. The input shape of the attention module is BxCx1xW or BxCxHx1 whilst that of the original axial attention module is BxCxHxW.  This will result in different ii) **Computation complexity,** the complexity of the Axial attention is O(N^1.5), N is a number of input features (tokens) and the proposed squeeze-enhanced Axial attention achieves linear complexity O(N), which is important for mobile applications. iii) **Special design in detail enhancement kernel.** To enhance the details information, we design a parallel convolution path as detail enhancement kernel. To further reduce the computation burden, we reuse the generated Q, K, V in attention, and concatenate them together as feature expansion in the detail enhancement kernel module. iv) **Better performance.** As shown in Table 4 (page 8) and Table 12 (page 16, appendix), the proposed squeeze-enhanced Axial attention achieves a better trade-off between accuracy and latency than the original axial attention.
>
> **Q3: Difference between the proposed architecture and TopFormer.**
>
> Figure 2 shows the schematic illustration of our whole network architecture. Apart from the attention block (Seaformer layer v.s. global attention layer), there are some significant differences between the proposed architecture and TopFormer.
> 1) **The usage of attention layer,** thanks to the linear complexity of the proposed SEA attention, we are able to use the Seaformer layer as the building unit to form the serval stages of the backbone per se rather than adding one old-school attention layer on the top of the backbone (TopFormer), clearly showing our superior cost-effectiveness design, especially on the higher-resolution semantic segmentation task.
> 2) **The way to fuse features,** we take a simpler way to gradually fuse the features from the shallow to deep layer for producing high-resolution rich-semantics features whilst TopFormer performs a complicated pyramidal network design: use the semantics injection module to fuse the output of different layers and then add the features from different stages together.
> 3) We will add the discussion in the revision to make it even clear.

---

> ### Author Response · Authors · 2022-12-05
> **Request for feedback on the rebuttal**
>
> Dear Reviewer wqtJ,
>
> We appreciate all the reviewing time and effort. With our best appreciation, we have made the revised paper and the response in detail to each individual comment. While we consider this could have addressed all the concerns raised hopefully, it is most critical that the reviewer can kindly read our response and tell us how the issues have been addressed and if any concerns are left to be addressed. We would take all the comments/suggestions as carefully as possible and address them with our best efforts. Many thanks for every effort the reviewer made and will make on our work.
>
> Best wishes,
> Authors

---

### Official Review · Reviewer_sRfJ · 2022-10-25

**Confidence:** 3
**Correctness:** 3
**Technical Novelty And Significance:** 2
**Empirical Novelty And Significance:** 3
**Recommendation:** 8

**Clarity, Quality, Novelty And Reproducibility:**

The paper is written very clearly with enough empirical evidence to support their claims. The work is based on axial attention, but there is enough originality to the method from an architecture design perspective.

**Strength And Weaknesses:**

Strength:

* The paper presents a new architecture for mobile semantic segmentation using ViT. Such dense prediction task is very challenging, and usually requires global attention blocks in ViTs to model long range dependencies. The paper presented an efficient attention block utilizing axial attention to tackle segmentation task.

* The paper presented solid numbers on 3 different segmentation datasets: ADE20K, Pascal Context, and COCO-Stuff. They also showed promising results on ImageNet for the image classification task.

Weaknesses:

* The paper proposed an efficient model for segmentation, but they didn't show any results on CityScapes to show the efficiency on dataset with large input size.
* The paper forgot to compare to some of recent work (e.g. EfficientFormer [1]).
* The paper listed a whole subsection (Shared STEM) in the method section which can imply it is a new work, instead it is based on previous work.

[1] Li, Y., Yuan, G., Wen, Y., Hu, E., Evangelidis, G., Tulyakov, S., Wang, Y. and Ren, J., 2022. EfficientFormer: Vision Transformers at MobileNet Speed. arXiv preprint arXiv:2206.01191.


**Summary Of The Paper:**

The paper presented a new mobile ViT architecture (SeaFormer) for semantic segmentation. The authors improved the computational cost of ViT architecture by proposing an efficient attention block, and light segmentation head. They showed solid results on ADE20K, Pascal Context, and COCO-stuff segmentation datasets.

Post rebuttal:

Thanks for clarifying the cityscapes results, and ImageNet. My rating will stay the same.

**Summary Of The Review:**

The paper presented SeaFormer: a new mobile-ViT model for semantic segmentation. The authors proposed an efficient architecture that model spatial and global information utilizing convnets, and an efficient axial attention block. Empirically, they showed solid results on ADE20K, Pascal Context, and COCO-stuff. This work benefit the effort to push ViT based models to be more mobile friendly.

---

> ### Author Response · Authors · 2022-11-16
> **Response to Reviewer sRfJ**
>
> Thanks for the positive and detailed review as well as the suggestions for improvement. We would like to reply to the comments as follows:
>
> **Q1:Cityscapes results.**
>
> A: The results on Cityscapes were included in the supplementary materials. We now put Cityscapes results into the main paper in our revised version. For convenience, we paste the results of Cityscape below. The experimental results demonstrate that the proposed SEA attention also achieves a good trade-off between segmentation accuracy and latency when processing high-resolution input images.
> | Method | Backbone|FLOPs|mIoU(val)|mIoU(test)|Latency|
> |----------|-------------|-|-|-|-|
> |FCN |MobileNetV2| 317G| 61.5| - |24190ms|
> |PSPNet |MobileNetV2| 423G |70.2 |- |31440ms|
> |SegFormer(h)| MiT-B0 |17.7G |71.9 |- |1586ms|
> |SegFormer(f)| MiT-B0 |125.5G |76.2 |- |11030ms|
> |L-ASPP |MobileNetV2| 12.6G |72.7 |- |887ms|
> |LR-ASPP |MobileNetV3-L| 9.7G |72.4 |72.6 |660ms|
> |LR-ASPP |MobileNetV3-S| 2.9G |68.4 |69.4 |211ms|
> |Simple Head(h)| TopFormer-B| 2.7G |70.7 |- |173ms|
> |Simple Head(f)| TopFormer-B| 11.2G |75.0 |75.0| 749ms|
> |Light Head(h) |Sea-Former-S |2.0G |70.7| 71.0 |129ms|
> |Light Head(f) |Sea-Former-S |8.0G |76.1 |75.9 |518ms|
> |Light Head(h) |Sea-Former-B |3.4G |72.2 |72.5 |205ms|
> |Light Head(f)| Sea-Former-B |13.7G |77.7 |77.5 |821ms|
>
> **Q2:Recent work comparison.**
>
> A: Thanks for the reminder. The results are as follows and we have added EfficientFormer into the revised version. We can see that SeaFormer outperforms these efficient approaches including EfficientFormer with comparable FLOPs and lower latency.
> |Method |Params| FLOPs |Top1| Latency|
> |---------|---------|---------|------|----------|
> |MobileNetV3-Small |2.9M| 0.1G |67.4| 5ms|
> |__SeaFormer-T__| 1.8M |0.1G |67.9| 7ms|
> |MobileViT-XXS |1.3M |0.4G |69.0 |24ms|
> |MobileViTv2-0.50| 1.4M |0.5G |70.2| 32ms|
> |MobileOne-S0*| 2.1M |0.3G |71.4 |14ms|
> |MobileNetV2 |3.4M| 0.3G| 72.0 |17ms|
> |Mobile-Former96| 4.8M |0.1G |72.8 |31ms|
> |__SeaFormer-S__|4.1M |0.2G |73.3 |12ms|
> |EdgeViT-XXS| 4.1M |0.6G |74.4 |71ms|
> |LVT |5.5M |0.9G| 74.8 |97ms|
> |MobileViT-XS |2.3M| 0.9G |74.8 |54ms|
> |MobileNetV3-Large| 5.4M |0.2G |75.2 |16ms|
> |Mobile-Former151| 7.7M |0.2G |75.2 |42ms|
> |MobileViTv2-0.75| 2.9M |1.0G |75.6| 68ms|
> |MobileOne-S1*| 4.8M |0.8G |75.9| 40ms|
> |__SeaFormer-B__|8.7M |0.3G |76.0 |20ms|
> |MobileOne-S2*| 7.8M |1.3G |77.4 |63ms|
> |EdgeViT-XS| 6.8M |1.1G |77.5 |124ms|
> |MobileViTv2-1.00| 4.9M |1.8G |78.1 |115ms|
> |MobileOne-S3*| 10.1M |1.9G |78.1 |91ms|
> |MobileViT-S |5.6M |1.8G |78.4 |88ms|
> |EfficientNet-B1 |7.8M |0.7G |79.1 |61ms|
> |EfficientFormer-L1|12.3M |1.3G |79.2| 94ms|
> |Mobile-Former508| 14.8M |0.5G |79.3 |102ms|
> |MobileOne-S4*| 14.8M |3.0G |79.4 |143ms|
> |__SeaFormer-L__| 14.0M |1.2G |79.9 |61ms|
>
> **Q3:Subsection introduction of Shared STEM.**
>
> A: Thanks for the reminder. The introduction of shared STEM is used to make the structure of the paper more complete. As we stated in the paper "For a fair comparison, we follow TopFormer to design STEM". We have revised this part in the revision to make it even clear.

---

### Official Review · Reviewer_xU22 · 2022-10-25

**Confidence:** 4
**Correctness:** 3
**Technical Novelty And Significance:** 2
**Empirical Novelty And Significance:** 2
**Recommendation:** 5

**Clarity, Quality, Novelty And Reproducibility:**

The paper is fairly easy to follow, although the writing could be slightly improved in some parts. I think the experiment clearly highlight some interesting aspects, but I find it hard to get actual insights into the proposed attention variant. Since the proposed improvement is somewhat incremental and we don't gain interesting insights, the whole approach feels a little ad hoc and more like and engineering paper, especially since some of the more interesting experiments are pushed out to the supplementary for some reason. I do estimate that the results should be reproducible, given that the authors already now uploaded code to some anonymous repository, thus seemingly willing to release the code.

**Strength And Weaknesses:**

Strengths:
- The results show that the overall architecture is indeed somewhat better and faster in the evaluated settings, even when actually running on a mobile device.
- The proposed architecture, specifically the new attention block does seem to have some novelty, however, it is fairly small incremental when comparing it to the axial attention.

Weaknesses:
- I'm mainly a bit surprised by some of the results. When looking at table 4, the clear take-away message is that the newly proposed attention is better than all other attentions. I'm assuming that the other attention mechanisms have been adjusted in such a way to keep the FLOP count roughly consistent, which could be a reasonable explanation why they don't outperform this fairly strong approximation of the vanilla global attention. But I find the investigation of this somewhat lacking. Does this now mean that we should start to use the new attention mechanism everywhere, by simply scaling it up to match the FLOP counts of the original global attention? Or is there some limit to this? Given that the main contribution of this paper is the squeeze-enhance axial attention, I would like to gain some more insights into why it seemingly works better than the global attention. However, I can't really gain these insights from the paper.
- Most of the actually interesting results are in the supplementary and I don't understand why the authors chose to write the paper like this. The supplementary actually has results on cityscapes, the only dataset used for experiments with a proper private test set. Furthermore it contains a more detailed analysis of the runtime, which is one of the main motivations for the whole idea of the paper. And finally, it also contains results of experiments with the proposed attention block in a SWIN former. I think this experiment is crucial to show that the proposed novelty actually has some form of generality. However, none of these results are even mentioned in the main paper, I don't really understand why it would be written in this way.


**Summary Of The Paper:**

The paper present an incremental improvement of axial attention. Instead of performing attention across the two axes for all pixels, feature maps are collapsed along the horizontal and vertical axes into two vectors. Self-attention is performed on these and the updated vectors can then be distributed across the original feature map at the input resolution. An additional detail enhancement block provides the original context of the feature map yielding the final output of a squeeze-enhance axial attention block. It is combined with a backbone (called the STEM) relying on MobileNet V2 blocks and some fusion modules which provide higher resolution information. The resulting SeaFormer architecture is evaluated for semantic segmentation and image classification, where there is a strong emphasis on the inference speed of the overall network, including when measured on mobile devices.

**Summary Of The Review:**

There are some compelling experiments show that some interesting aspects of the proposed attention mechanism exist, however, I would like to actually get a better understanding why it apparently works so much better and what the limitations are. Overall I'm thus a bit torn when it comes to accepting the paper or rejecting it. However, depending on the other reviews and rebuttal, I'm willing to improve my rating.

---

> ### Author Response · Authors · 2022-11-17
> **Response to Reviewer xU22**
>
> Thanks for the insightful and detailed review as well as the suggestions for improvement. We would like to reply to the comments as follows:
>
> **Q1: Insights.**
>
> 1) The mobile-friendly segmentation is deeply related to the industrial application on edge computation platforms, while few academic attempts are made to meet the requirement of the industry. **The intuition behind our work is that we need to solve the dilemma of high-resolution computation for pixel-wise segmentation task and low latency requirement on mobile devices.** This is non-trivial and we thus propose a generic attention block characterized by the formulation of squeeze Axial and spatial enhancement and further create a family of backbone architectures with superior cost-effectiveness to beat state-of-the-art alternatives.
> 2) **We strongly advocate that our approach per se significantly revolutionizes mobile semantic segmentation**, especially for the high-resolution per-pixel semantic segmentation task. The other virtue is that our model also tackles the image classification problem, demonstrating the potential of serving as a versatile mobile-friendly backbone.
> 3) Regarding performance, we beat both the state-of-the-art mobile-friendly rivals and state-of-the-art Transformer-based counterparts with better performance and lower latency without bells and whistles.
> 4) Given its conceptual simplicity, versatility, and fast speed, our method can serve as a strong baseline and inspire further studies that consider mobile-friendly methods on edge computation platforms. We believe our work can lead to expected and unexpected innovations in both academia and industry.
>
> **Q2: Experimental setting of Table 4.**
>
> Based on the Reviewer xU22's comments, we have moved Table 4(a) into the section Appendix and put the Cityscapes, latency statistics, and Swin Transformer backbone experiments into the main paper in our revised version.\
> For a fair comparison, we adjust the length of the feature channel when applying different attention modules to keep the FLOPs roughly aligned and then compute their performance and latency on ADE20K. We have stated it in the subsection Effective and efficiency of SEA attention (page 15, appendix).\
> We have also conducted experiments to verify the effectiveness of the proposed SEA attention in the subsection Comparison with different self-attention modules (page 8) based on Swin Transformer architecture and we set the same model architecture configurations for all attention variants.
>
> **Q3: Better performance than global attention?**
>
> 1) In terms of the single attention block, our SEA attention consists of squeeze axial attention and detail enhancement kernel (Figure 3). From our analysis (subsection Detail enhancement kernel, page 5), results of line 2 in Table 3 (page 8) and line 4 in Table 12 (page 16, appendix), squeeze axial attention without detail enhancement (Top1 66.3%, mIoU 33.5%) is indeed **inferior** to the global attention (Top1 66.7%, mIoU 34.2%) as it sacrifices the local details in the squeeze operation. This also motivates us to develop a parallel detail enhancement kernel. Enhancing global semantic features of the squeeze axial attention with local details optimizes the feature extraction capability of the Transformer block, as illustrated in line 2 and line 3&4&5&6 of Table 3 (page 8). The qualitative results in Figure 5 (page 17, appendix) also demonstrate that squeeze axial attention with detail enhancement activates the semantic local region accurately, which is particularly significant in the dense prediction task (e.g. semantic segmentation).
> 2) On the other hand, our contribution not only lies in the proposed single attention module, **but more importantly** in the whole backbone architecture design. We strongly advocate that **the latter** very much matters for the mobile-friendly task. Designing a single attention module and integrating it into existing backbone architectures is actually **cheap**. The importance of efficiency is usually underscored by the fact that the capabilities of many mobile and embedded devices are limited.
>
> **Q4: Limitation of SEA attention.**
>
> SEA attention is designed for specific application scenarios and we have verified its effectiveness for vision tasks including semantic segmentation,  image classification, and object detection, on the arm-based devices. It may not achieve superior performance in some other scenarios. Also, we are committed to improving the inference efficiency, the number of parameters could be a problem if you scale up to an extremely large model size.
>
> **Q5: Some results in supplementary are more crucial.**
>
> Many thanks for the advice and the appreciation of our experimental evaluation. We have put these experiment results (e.g. Cityscapes, latency statistics, and Swin Transformer backbone) into the main paper in our revised version.

---

### Author Response · Authors · 2022-11-17
**General Response**

We sincerely appreciate all reviewers’ time and efforts in reviewing our paper. We truly thank all for the insightful and constructive suggestions, which helped further improve our paper.

We addressed the reviewers’ concerns and suggestions in our reviewer-specific responses. Based on the valuable suggestions and reviews of the reviewers, we revised our manuscript, and the modifications in the revised version are summarized below:

* Moved CityScapes results, latency statistics, and Swin Transformer architecture experiments to the main paper.

* Added “EfficientFormer” in subsection Image classification to compare with recent work.

* Modified the introduction of shared STEM.

* Modified the name of fusion in Figure 3 to avoid confusion.

* Modified the statement of state-of-the-art.

* Updated the bib file.

* Separated  Table 2 and Table 4.

* Revised the order of Table 3 and Table 4.

* Added discussion on a generic block.

We are confident that our response should have cleared the air, and we are happy to answer any additional questions and provide more information.

We really thank all reviewers’ time and efforts again.

Best wishes,

Authors

---

### Decision · Program_Chairs · 2023-01-20

**Decision:**

Accept: poster

**Justification For Why Not Higher Score:**

The novelty is more a clever engineering design.

**Justification For Why Not Lower Score:**

It is worth publishing something that is an engineering design.

**Metareview: Summary, Strengths And Weaknesses:**

The reviewers agree that the proposed method for semantic segmentation using ViT speeds ups inference in mobile devices. They agree also that the proposed novelty of axial attention blocks instead of global attention is incremental. However, the results are solid on the main semantic segmentation datasets showing superior results than TopFormer. They achieve that by adding a design for detail enhancement using depth-wise convolution to aggregate local spatial details.

The main weakness is that the novelty is more a clever engineering design with existing blocks. However, I consider that as a fair novelty given that the paper shows improved results, is useful and beats state of the art. There was other weaknesses such as comparing with other baselines, concerns on some of the experiments, more details needed in state of the art review, improved citation style needed, and notation.

The authors addressed many of these concerns such:
- Moved CityScapes results, latency statistics, and Swin Transformer architecture experiments to the main paper.
- Added “EfficientFormer” in subsection Image classification to compare with recent work.
- Modified the introduction of shared STEM.
- Modified the name of fusion in Figure 3 to avoid confusion.
- Modified the statement of state-of-the-art.
- Updated the bib file.
- Separated Table 2 and Table 4.
- Revised the order of Table 3 and Table 4.
- Added discussion on a generic block.

Accordingly, I suggest accepting this article.

**Note From Pc:**

if the above contains the word "oral" or "spotlight" please see: "oral" presentation means -> notable-top-5% and "spotlight" means -> notable-top-25%. As stated in our emails, we are disassociating presentation type from AC recommendations